# T2*-Mapping of Knee Cartilage in Response to Mechanical Loading in Alpine Skiing: A Feasibility Study

**DOI:** 10.3390/diagnostics12061391

**Published:** 2022-06-04

**Authors:** Uwe Schütz, Thomas Martensen, Sebastian Kleiner, Jens Dreyhaupt, Martin Wegener, Hans-Joachim Wilke, Meinrad Beer

**Affiliations:** 1Department of Diagnostic and Interventional Radiology, University Hospital Ulm, 89081 Ulm, Germany; thomas-martensen@web.de (T.M.); martin.wegener@uniklinik-ulm.de (M.W.); meinrad.beer@uni-ulm.de (M.B.); 2Department of Nuclear Medicine, Technical University Munich, Klinikum Rechts der Isar, 81675 München, Germany; sebastian.kleiner@mri.tum.de; 3Department of Biometrics, University Hospital Ulm, 89081 Ulm, Germany; jens.dreyhaupt@uni-ulm.de; 4Trauma Research Centre, Institute of Orthopaedic Research and Biomechanics, Ulm University, 89081 Ulm, Germany; hans-joachim.wilke@uni-ulm.de

**Keywords:** T2-mapping, cartilage, osteoarthritis, knee, ski, MRI

## Abstract

Purpose: This study intends to establish a study protocol for the quantitative magnetic resonance imaging (qMRI) measurement of biochemical changes in knee cartilage induced by mechanical stress during alpine skiing with the implementation of new spring-loaded ski binding. Methods: The MRI-knee-scans (T2*-mapping) of four skiers using a conventional and a spring-loaded ski binding system, alternately, were acquired before and after 1 h/4 h of exposure to alpine skiing. Intrachondral T2* analysis on 60 defined regions of interest in the femorotibial knee joint (FTJ) was conducted. Intra- and interobserver variability and relative changes in the cartilage T2* signal and thickness were calculated. Results: A relevant decrease in the T2* time after 4 h of alpine skiing could be detected at the majority of measurement times. After overnight recovery, the T2* time increased above baseline. Although, the total T2* signal in the superficial cartilage layers was higher than that in the lower ones, no differences between the layers in the T2* changes could be detected. The central and posterior cartilage zones of the FTJ responded with a stronger T2* alteration than the anterior zones. Conclusions: For the first time, a quantitative MRI study setting could be established to detect early knee cartilage reaction due to alpine skiing. Relevant changes in the T2* time and thus in the intrachondral collagen microstructure and the free water content were observed.

## 1. Introduction

In alpine skiing, high peak loads affect the knee joint, and this is confirmed by the comparatively high rates of knee injury in this popular sport [1]. Skiing is practiced by all age groups, with the over-60s making up about 20% of skiers [2]. The uptake rates in the bone scintigraphy (using the Tc-99 m MDP bone scans) for the knee joints during the active racing period were significantly higher than those during the inactive period. This indicates an increased risk of damage to the knee cartilage and the development of osteoarthritis (OA). Based on these figures, it can be assumed that many individuals who are already affected by OA regularly engage in alpine skiing.

The two major macromolecular components of the extracellular cartilage matrix, Type-II collagen and proteoglycans (PG), are responsible for its biomechanical resilience, with collagen providing elastic properties [3] and PG providing viscoelastic properties [4]. Water occupies most of the interfibrillar extracellular matrix, approximately 70% of which is free to move when loaded by compressive forces [5]. OA begins with an impaired balance of cartilage metabolism, with changes in the microstructure of collagen fibers, a loss of PG, and an increase in the free water content. Increasing OA is accompanied by a progressive reduction in the free intrachondral water content. These changes lead to irreversible damage to the cartilage during the course of the disease [6].

Quantitative magnetic resonance imaging (qMRI) offers the possibility of detecting changes in the chondral biochemical microstructure before morphological changes have occurred and can therefore be used as a non-invasive biomarker of cartilage degeneration [7]. Numerous studies have examined the changes in cartilage under biomechanical load during running using qMRI (dGEMRIC, T1rho-, T2/T2*-mapping, etc.) [5,8,9]. Among these, T2*-mapping is suitable for detecting the proportion of intrachondral free water and the collagen fiber structure, thus detecting the early stages of OA [10]. The comparative advantages of T2*-mapping are the short acquisition time, the avoidance of contrast, the high spatial resolution, and the possibility of isotropic three-dimensional reconstruction [11]. Previous joint studies in alpine skiing are limited to the evaluation of elderly individuals with unilateral total knee replacement in terms of perceived pain, knee function, muscle mass, and effort [12]. Due to logistical, temporal, and technical peculiarities, scientific work investigating the relationship between alpine skiing and the microstructure of cartilage using qMRI has not yet been carried out.

The aim of this preliminary feasibility study is to establish a suitable examination protocol to measure and analyze any relevant intrachondral changes in the femorotibial (FTJ) and femoropatellar joint (FPJ) by means of mechanical stress during alpine skiing using qMRI (T2*-mapping). The changes were examined on a conventional ski binding without suspension and on a new spring-mounted ski binding, which can be mounted between any alpine ski model and the ski binding. The damping plate consists of duralumin with pressed-in plain bearing sleeves and an inserted leaf spring made of austenitic stainless chromium-nickel steel (Appendix A). According to the manufacturer, the damping plate absorbs up to 40% of the impact load—for example, on icy ski slopes or through transverse grooves [13]. By reducing the impact load on the joints, the damping plate is especially considered for skiers with pre-existing OA.

## 2. Materials and Methods

For the study carried out in the ski resort of Lech/Zürs (Austria), four experienced male alpine skiers from a South German ski club were randomly selected. The subjects were 18 (subject 1), 22 (subject 3), 32 (subject 4), and 58 (subject 5) years old and had a BMI of 22.0, 28.8, 22.6, and 24.8 kg/m², respectively. The study was approved by the Ethics Committee of “blinded”.

Prior to enrollment, all of the subjects underwent an orthopedic-clinical basic examination and an initial morphologically oriented MRI of both knee joints. The clinical status and alignment of the lower extremities and their knee joints showed no relevant abnormalities or pathologies. This excluded arthralgia (no pressure pain, negative meniscus test mix [14]), instabilities/laxities (testing of the ligaments: negative Lachmann, (reversed) pivot shift, Lever Sign test [15]), dysfunctions (stable patellar movement), and inflammation (no swelling, warming, effusion). Morphological (PDfs: proton density fat saturated) and water-sensitive (TIRM: turbo inversion recovery magnitude) MRI sequences (Table 1) were obtained and subsequently assessed for the detection of effusion/edema and (osteo-) chondral lesions in the knee using the MRI-modified Outerbridge grading system [16]. This excluded relevant chondropathies (higher Outerbridge grade 1, [17]), meniscal lesions with surface involvement (higher grade 2, [18]), ligamentopathies, and inflammation in all subjects.

The qMRI study took place on two consecutive weekends. The test subjects chose a slope profile with similar levels of difficulty on both weekends (category blue or red). Due to foehn storms (high speeds of the warm, fall wind on the leeward side of the mountains), only about 50% of the ski lifts in the ski resort were open on the first weekend. On the second weekend, about 90% of the lifts were open, which made the waiting times of the subjects at the lifts correspondingly shorter. On the first weekend, the mean air temperature was 6.3 °C on the day of skiing; on the second, it was −0.7 °C. There were softer snow conditions on the slopes on the first weekend (data acquisition: Central Institute for Meteorology and Geodynamics Austria).

For qMRI data acquisition, a 1.5 T MRI (Magnetom Aera, Siemens Ltd., Siemens Healthcare GmbH, 91052 Erlangen, Germany) was available at the outpatient office Lech, which was at a distance of 200 m from the ski slope. The qMRI scans with T2*-mapping (Table 1: Syngo^TM^ MapIt FLASH T2*-GRE) using a dedicated, table fixed, eight-channel knee coil were performed for both knee joints of the subjects one day before (baseline: I-t0), immediately after 1 h (I-t1) and 4 h (I-t2) of loading by alpine skiing, and after recovery the following morning. Subjects 1 and 2 used a conventional ski binding without suspension on the first weekend (I), and the damped ski binding system on the second weekend (II). Subjects 3 and 4 used the binding systems in reverse order. On the second weekend, the same was repeated with the changed binding systems (baseline: II-t0, II-t1, II-t2). A blinding of the subjects regarding the ski binding systems could not be realized. For the quantitative biochemical cartilage analysis, T2* relaxation times were obtained from online reconstructed T2* maps by using a pixelwise, mono-exponential nonnegative least-squares-fit analysis (Syngo^TM^ MapIt; Siemens Ltd.) [19]. The mean latency between the end of the skiing and the start of the T2* measurements was less than 10 min, and the mean MR scan duration for both knee joints was 38 min.

The use of qMRI techniques for the structural analysis of (knee) articular cartilage requires zonal differentiation in order to detect specific cartilage degradation patterns and analyze causal relationships [20]. This is in line with our approach. The mean cartilage height (Ht), the T2* time per region of interest (ROI), and the mean ROI size, calculated by the areal dimension and the number of pixels per ROI, were measured and documented on an MR workstation (Syngo^TM^ MapIt, Siemens Ltd.) by a trained scientist in the FTJ and FPJ. For this purpose, a total of 18 ROIs were defined on each knee for the cartilage layers (12 in the FTJ and 6 in the retropatellar cartilage layer) based on the methodology of the T2*-mapping of the TransEurope FootRace (TEFR) project [9], first described by Mamisch et al. [21]. As illustrated in Figure 1B–D, the ROIs were manually drawn on the slices in a way that covers nearly the entire cartilage areas. Care was taken to avoid the subchondral bone or joint fluid and to set the ROIs in the exact same positions at every examination. Table 2 shows the nomenclature of specific cartilage ROI areas. In total, four slices were implemented per FTJ (Figure 1A), and two slices were implemented per FPJ (=60 ROIs). At five measurement times, a total of 600 ROIs per subject and 2,400 ROIs for the overall study had to be created. For each ROI, the average T2*, number of pixels, and Ht were calculated from the two parallel slices and taken for further analysis. To determine the mean T2* for each layer, zone, and cartilage segment (Table 2), the mean T2* values of the specific area were pooled and calculated with regard to the ROI sizes. The time required for the cartilage T2* and Ht analysis was nearly 45 min for each side.

**Statistics and Testing.** For the determination of the intra- and interobserver variability of the ROI sizes and the mean T2* and Ht values, the data of a randomly selected subject were again evaluated by the same scientific staff member after 6 months and additionally by a specialist in radiology. They were supervised by two radiologists with a special interest in musculoskeletal imaging and 15–25 years of experience. The intra- and inter-class correlation coefficient (ICC) was calculated for each ROI (n = 140) [22], and Bland–Altman plots were created to visualize the match for the T2* time, the number of pixels, and the Ht (95% limits of agreement (LOA): mean difference ± 1.96 standard deviation (SD)) [23].

For the data documentation, statistical and descriptive analyses and graphical presentations using Office-Excel^TM^ (release-1812, 2016, Microsoft Inc., Microsoft Corporation, Redmond WA 98052-6399, USA), SPSS^TM^ (release-25.0, 2017, IBM^TM^-Statistics), and SigmaPlot^TM^ (release-12.5., 2011, Systat Inc., Systat Software GmbH, 40699 Erkrath, Germany) were utilized, respectively. It is known from previous studies that absolute intrachondral T2/T2* and cartilage thickness values show joint related intraindividual values due to multiple influencing factors such as gender, age, weight, activities of daily living (ADL), joint anatomy, alignment, etc. [24,25,26]. Therefore, the calculated and graphed target parameters were the relative changes of T2* and Ht compared to the baseline I and II, respectively. Due to the small number of subjects, no statistical analyses were done.

## 3. Results

For the mean T2* time per ROI, the Intra-ICC was 0.95 (confidence interval (CI) 0.93–0.96) and the Inter-ICC was 0.92 (CI 0.89–0.94), with segmental SDs of 0.92 ms and 1.15 ms, respectively; the corresponding LOAs are shown in Figure 2. For the mean number of pixels per ROI, the Intra-ICC was 0.96 (CI 0.95–0.97) and the Inter-ICC was 0.93 (CI 0.90–0.95); the segmental SDs were 14.4 px and 16.9 px, respectively (for the LOAs, see Figure 2). For the mean Ht, an Intra-ICC of 0.92 (CI 0.88–9.4) was determined, with a segmental SD of 0.21 mm (for the LOA, see Figure 2).

At baseline, the mean measured Ht in the FPJ was greater compared to that in the FTJ. Compared to baseline, the after-load measurements in the FPJ and FTJ showed a slightly lower Ht in the majority of the measurement series (cartilage segments, Appendix A). The SD of the measured changes in cartilage height Ht was regularly below 0.22 mm (Appendix A).

Figure 3 exemplifies the change in the measured T2* time of the color-coded T2* map on the lateral femoral articular cartilage. Here, after a loading time of 4 h (I-t2), especially in the anterior part of the femoral cartilage, a clear decrease in the green color signal is visible for the decrease in T2* time. In the recovery measurement, the T2* time or the green color signal of the cartilage recognizably increases again.

The segment-related relative changes in the T2* time are graphically shown in Figure 4a–c (for the corresponding mean T2* values, see Appendix A). The two youngest subjects had lower mean T2* values than the other two subjects at baseline in both the FTJ and retropatellar. The mean T2* times were 28.6 ms (SD 2.9 ms) for the lateral femoral, 19.6 ms (SD 1.5 ms) for the lateral tibial, 26.9 ms (SD 2.1 ms) for the medial femoral, and 23.3 ms (SD 2.2 ms) for the medial tibial; for the retropatellar, it was 26.6 ms (SD 3.2 ms). Side-differences in the subjects with regard to the T2* values could not be detected (Appendix A). Compared to the deep cartilage layers, the superficial layers across all of the knee segments and test persons showed a significantly higher T2* signal at all times of measurement (Figure 5); the mean difference at the baseline measurement in the lateral FTJ was 7.9 ms (SD 2.7 ms), in the medial FTJ it was 9.2 ms (SD: 3.5 ms), and in the retropatellar it was 10.2 ms (SD: 2.1 ms). The zonal T2* comparison at baseline showed no relevant differences in the mean values due to the comparatively high SD (Appendix A).

In all of the subjects, relevant changes in T2* time could be detected under load in both knee joints compared to the baseline (Figure 4a–c). However, in comparison, increases and decreases in T2* time were measurable in the individual subjects. No relevant mean trend could be evaluated for higher or lower T2* changes with respect to layers (Figure 5), zones (Figure 6), or segments or side differences. However, when looking at Figure 6, the middle and posterior zones of the femoral and tibial segments experience a higher T2* change than the anterior zones. The load-induced relative T2* changes also showed no trend-setting differences with respect to the knee joint compartments or regions. After 4 h of exposure, a decrease in T2* time compared to the baseline was observed in the majority of subjects in the FPJ and retropatellar. In this regard, however, no significant differences can be demonstrated between the measurements after 1 h or after 4 h of exposure.

At the recovery measurement, the T2* time in all four subjects was elevated in almost all FTJ segments but not in the retropatellar (Figure 4).

Skiing comfort was rated as high by all subjects for both the conventional ski binding and the damped binding systems. Due to the different weather conditions on both weekends, with much softer slope conditions on the first weekend, the subjective comparison of the ski binding systems for the subjects was only possible with difficulty. Regarding the T2* times, no relevant difference between the two attachment systems was detectable (Figure 4, Figure 5 and Figure 6).

## 4. Discussion

The authors present an application of qMRI for the biochemical evaluation of articular knee cartilage in alpine skiers, focusing on specific T2*-mapping for the first time. The location of the MRI nearby a ski slope provided optimal conditions, since the subjects could be examined immediately after skiing without any relevant loss of time, allowing for the minimization of recovery times.

The ICCINTRA and ICCINTER showed high agreement for the tested parameters T2*, ROI area, and Ht. What is decisive for the interpretation of the sufficient accuracy of a measuring method, however, is the SD or the LOA (=1.96 + SD).

**Cartilage Ht.** Compared to the baseline, a somewhat lower Ht was measured in the majority of the measurement series in the cartilage segments after ski load (Table 3). However, these measurements are not reliable because the intraobserver inaccuracy (SD and LOA, Figure 2) of the measurement method used is at least the same as the measured changes. The cartilage Ht measurement methodology should therefore be considered unsuitable after this feasibility study, as it is considered too imprecise, and any subsequent study should use other MR-based cartilage thickness or volumetric analysis methods that are more accurate and already validated [27]. With these, a comparison could then be possible with regard to the study results on cartilage Ht change during running, explaining the short-term load-induced cartilage Ht reduction, as in the orientation of the collagen fiber structure due to compression forces—especially in the transitional zone of the cartilage [5].

**Cartilage T2*.** In all of the subjects, the relevant cartilage T2* changes due to alpine skiing could be measured. Many studies on endurance running have shown higher load-induced changes for the superficial compared to the deep cartilage layers [25]. However, we could not prove such regional differences regarding T2* changes.

Some studies indicate that measured T2* values are directly dependent on the type of exercise and their specific cartilage loadings and therefore can lead to different T2/T2* maps [24]. Due to the recurrent knee flexion and/or prolonged downhill posture, alpine skiing has a different mechanical load on the knee compartments compared to running. Because of the movement pattern, skiing is more akin to squatting than running. When running, the compression force on the FTJ is about three times the body weight, but it is up to six times the body weight in squatting [28]. The compression force on the patella during running is about 5.6 times the body weight and up to 7.8 times the body weight in squatting [29]. As no T2 mapping studies have been performed so far for alpine skiing, the presented measurement results are not directly comparable. Regarding the literature on stair climbing, the superficial layers of the retropatellar lateral cartilage should undergo bigger T2 changes [8], which we could not regularly evaluate in any subject (Figure 6). We were unable to evaluate any tendency for higher retropatellar T2* changes compared to the FTJ segments, but the conspicuousness of the higher load-induced T2* changes in the middle and posterior femorotibial zones coincides with the T2 map observations under axial loading due to running [8]. Regarding the intra-rater values, the measured T2* changes are relevant and therefore load-induced; however, due to the small number of cases, no regional or ski-related trends or influencing factors could be evaluated.

At t2, a T2* decrease was observed in the majority of subjects in the FTJ and retropatellar. An initial intrachondral T2/T2* decrease after running has been observed for the knee joint and is described as acute cartilage reaction due to running impact [30]. Since there is a de facto shorter effective load due to waiting times at the lift and lift transport times, it seems reasonable to compare the results of this study with studies that investigated shorter loads. MR studies that examined the changes in intrachondral free water content after 30 min of running also showed an immediate T2 decrease [5,8,30,31], which is caused by the compression-induced mechanical displacement of free water from the cartilage [3,5]. Another reason for this signal decay is described by the increase in the anisotropy of superficial collagen fibers [30,32]. On the other hand, we partially observed T2* increases, which may be caused by PG degradation and the release of water molecules from molecular binding [9] or by the uptake of free water from the perichondral environment (edema theory) [33]. Therefore, a specific causal interpretation of our observed T2* changes remains open. Different mechanisms of the displacement of free intrachondral water [10], as well as the different regional and zonal reactions of collagen fiber anisotropy, seem to overlap [34].

In all of the segments, the intrachondrial T2* signal increased after overnight recovery in three of the four subjects. Other study results show an increase in the T2/T2* signal after the prolonged exposure to a marathon [35]. Increasing T2* values reflect free water dissociated from its chemical bonding to PG [10] and is related to PG content [34]. The initial stages of OA are also accompanied by microstructural changes, with an increase in the water content in the cartilage [10]. The increase in the T2* signal after exercise in our study can be interpreted either as an early sign of (reversible) cartilage degradation by transient changes in chondral homeostasis (the loss of structural anisotropy in the collagen matrix and a concomitant increase in free intrachondral water with decreasing PG content) [34] or as an excessive or compensatory uptake of free water into the cartilage [36].

**Limitations of the study.** The small number of subjects does not allow for statistical evaluation or testing for differences or influencing factors. Therefore, between the conventional ski binding system and the damped binding system, no significant differences in T2* times could be analyzed. Although the subjects chose a similar slope profile on both weekends, an influence of the individual driving style of the experienced skier on the load of the cartilage is conceivable.

Other possible influencing factors are the different climatic and snow conditions at both weekends, as well as the short but partly varying latency between the end of the loading and the start of the MR scan. During the unloading of the chondrocytes, a complete recovery of all structural deformation was observed after 30 min [37]. The volumetric MRI analyses of the ankle cartilage showed significant initial talar cartilage volume reduction, which was restored within 30 min [38]. In this respect, it must be assumed that the load-related initial decrease of the T2 signal already begins to decline with the end of the loading. The time interval of the T2 measurement from the end of the load is therefore decisive for these measurements—if one wants to detect the initial cartilage reaction. The transport of the subjects from the ski lift with a wheelchair directly into the MRI would, in a follow-up study, eliminate the unpredictable influencing factor of the very short but non-specific walking load after skiing with regard to T2 signaling.

Alternatively, the implementation of a T1rho-measurement would be conceivable. It detects structural changes in the PG and is considered by some authors to be more sensitive compared to T2/T2* measurements [39]. In addition, the accuracy of the established qMRI cartilage analyses is getting better as a result of the implementation of high-resolution MR protocols, so the use of appropriate systems for future comparable studies is recommended [40].

Age, activity level, and physical fitness [41] also have an impact on cartilage composition and physiology and thus on cartilage T2/T2*. While some authors assume that age does not have a relevant influence on the initial T2 response after running [5], regional differences, especially for the superficial [25] and for the deep cartilaginous layers [26], have been described. In the discussion of these inconsistent findings, it is argued that these may be due to the OA-related structural pathology, particularly cartilage damage, which becomes more common with age even when there are no radiographic signs of knee OA [42]. Therefore, the age-related cartilage T2 observed in earlier studies may be more likely to be due to cartilage pathology than to normal cartilage aging and may not be observed if the risk factors for knee OA are rigorously eliminated. However, this is exactly what Wirth et al. [26] did in their study, in which they detected age-related differences in the composition of the deep cartilaginous layer in the knee joint using T2 mapping. There could also be a positive effect of joint loading on the chondrocyte function associated with increased PG and collagen synthesis [43]. In elite runners and untrained volunteers, exercise increases PG content [44]. Increasing hydrostatic pressure upregulates PG and Type II collagen mRNA expression [45], and the de novo synthesis of PG will be initiated. So, due to the still-not-completely-clear literature review [5,26], the fact that one of the subjects was 26 to 40 years older than the other three, and the fitness level in general, the activity profile of the subjects in the run-up to the study must also be listed as a possible influencing factor on the detected initial T2 signal behavior. 

Therefore, the approach to a subsequent RCT study should be based on a significantly higher number of cases in order to be able to statistically verify the results or specifically investigate new protective binding systems. It would be best to homogenize influencing factors (climatically stable environmental conditions; comparable snow and slope conditions, e.g., by indoor/laboratory conditions; age-homogenized test subjects with a similar level of activity; defined skiing styles and profiles) and to optimize MR protocols regarding specificity (T1rho and T2 measurements) and accuracy (high-resolution qMRI imaging and possibly 3T).

As there are still no limit values defined, which allow for the differentiation of the initial intrachondral T2/T2* signal behavior after loading with respect to physiological and pathological cartilage reaction, and because of the recognition that recurrent cartilage load may also be a chondroprotective and regenerative stimulus for articular cartilage [32,35], direct observation of only the immediate cartilage response to skiing load may well be half the truth in terms of the overall effect. Therefore, such studies should include longer-term observation periods.

## 5. Conclusions

In conclusion, the additional logistical effort of the presented study setting with an MRI scanner within a large ski seems to be worthwhile, as reliable measurement results in T2*-mapping immediately after alpine skiing with corresponding load-induced T2* signal changes could be quantified and documented for the first time. After alpine skiing, the T2* times at most measurement points in the knee cartilage decrease immediately after exercise and increase beyond the baseline in 12–18 h. T2*-mapping can be used as a promising, non-invasive biomarker to detect early cartilage degradation on the knee joint due to biomechanical load during alpine skiing, and, with a sufficiently high number of cases under comparable environmental conditions, it may be a method for the evaluation of different knee joint cartilage loading responses in relation to external factors influencing alpine skiing (for example, different ski binding/buffering systems).

## Figures and Tables

**Figure 1 diagnostics-12-01391-f001:**
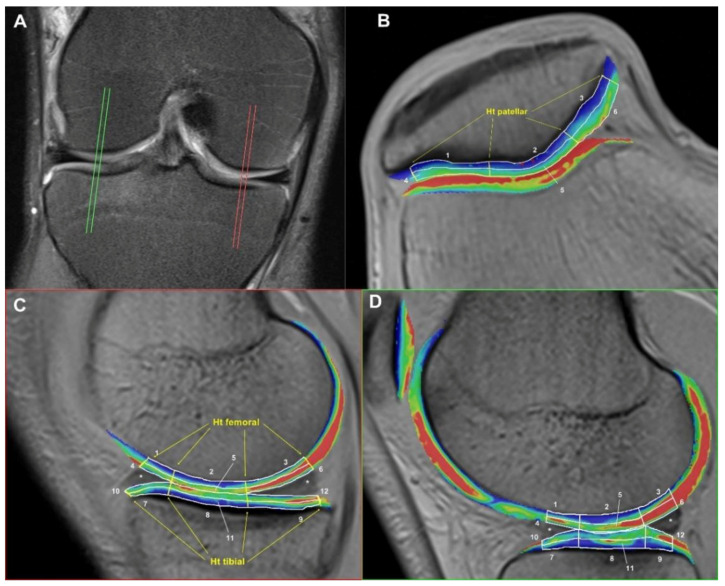
MR Image Postprocessing for the Quantification of Cartilage Ht and T2* (the numbered cartilage ROIs match to those, as explained in Table 2). (**A**): Four sagittal slices (two central layers each in the lateral and medial FTJ); (**B**): fused colored T2* maps (syngo™-MapIt fusion technique) of axial FLASH T2*w GRE in the FPJ with ROIs for patellar T2* and Ht measurement between the medial, central, and lateral zones; (**C**,**D**): fused colored T2* maps of sagittal FLASH T2*w GRE in the medial FTJ (**C**) and lateral FTJ (**D**) with ROIs for T2* and Ht measurement between the anterior, central, and posterior zones of the femoral and tibial cartilage segments. The coloring relatively reflects the intensity of the water signal or the water concentration, respectively, reaching from blue to green, yellow, and red. Blue: weak signal / low water content, red: strong signal / high water content.

**Figure 2 diagnostics-12-01391-f002:**
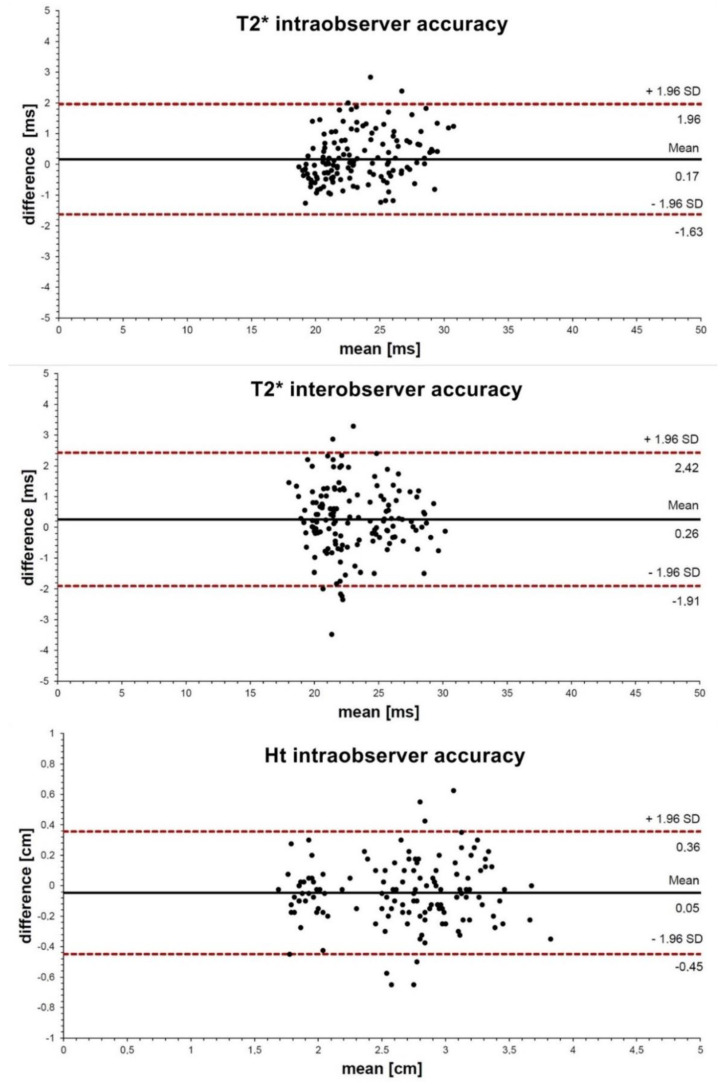
Bland–Altman Plot for Measured Cartilage T2* and Ht: intra- and interobserver accuracy (black line: mean), with 95% limits of agreement (LOA: dotted red lines: mean difference ± 1.96 SD).

**Figure 3 diagnostics-12-01391-f003:**
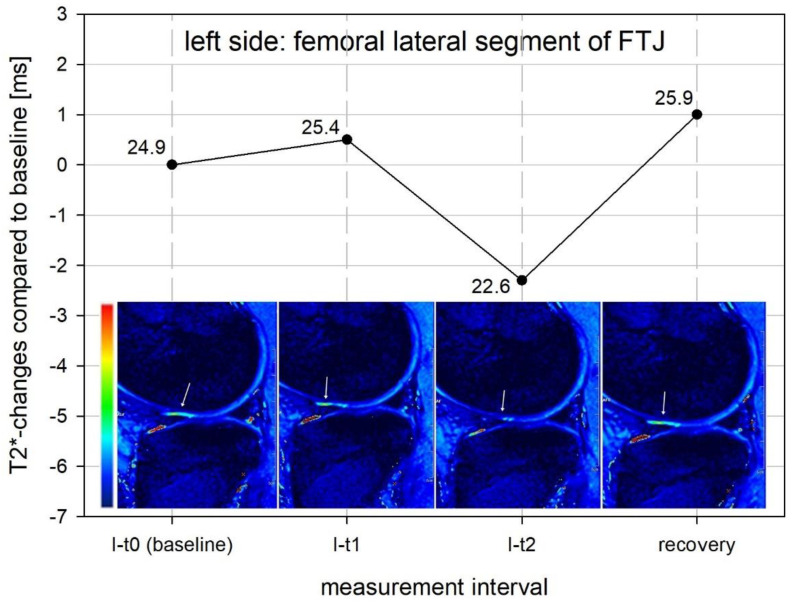
Color-Coded Mapping of T2*-Signaling in Comparison: decrease in green color signal indicating T2* decrease at I-t2, and renewed T2 increase at the recovery measurement corresponding to the green color signal increase.

**Figure 4 diagnostics-12-01391-f004:**
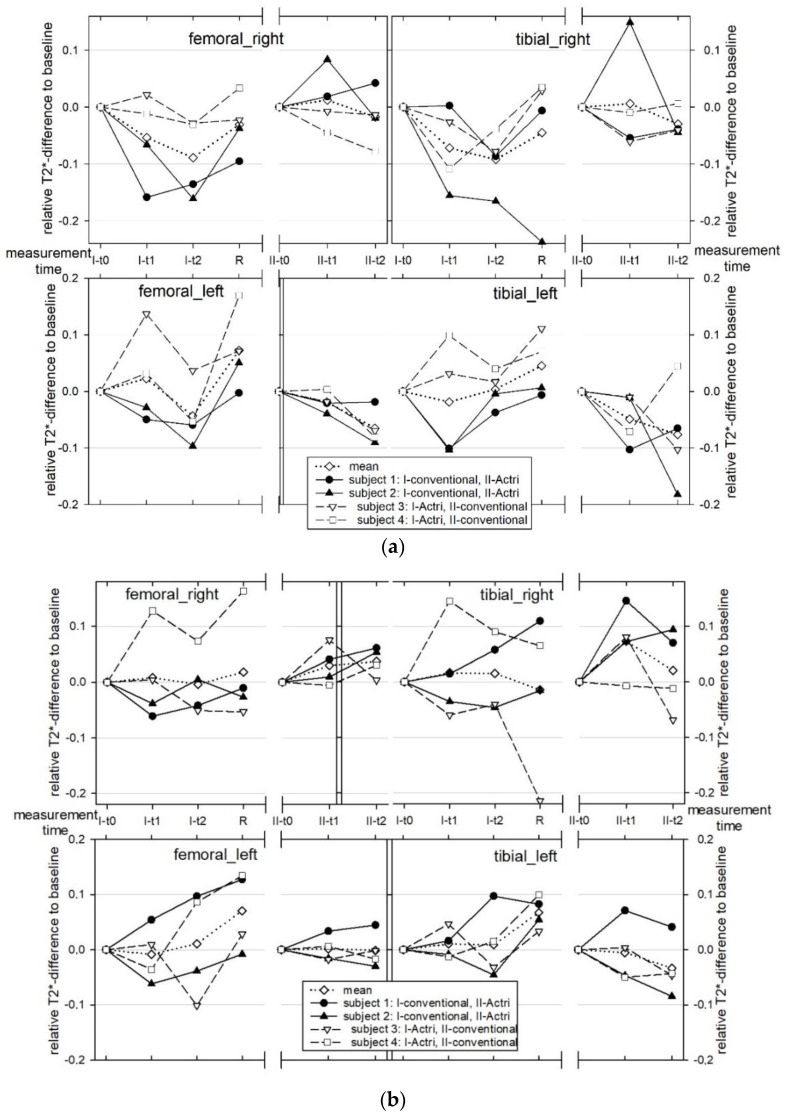
(**a**) Relative Changes of Mean Cartilage T2* in the Medial FTJ. (**b**) Relative Changes of Mean Cartilage T2* in the Lateral FTJ. (**c**) Relative Changes of Mean Cartilage T2* in the retropatellar.

**Figure 5 diagnostics-12-01391-f005:**
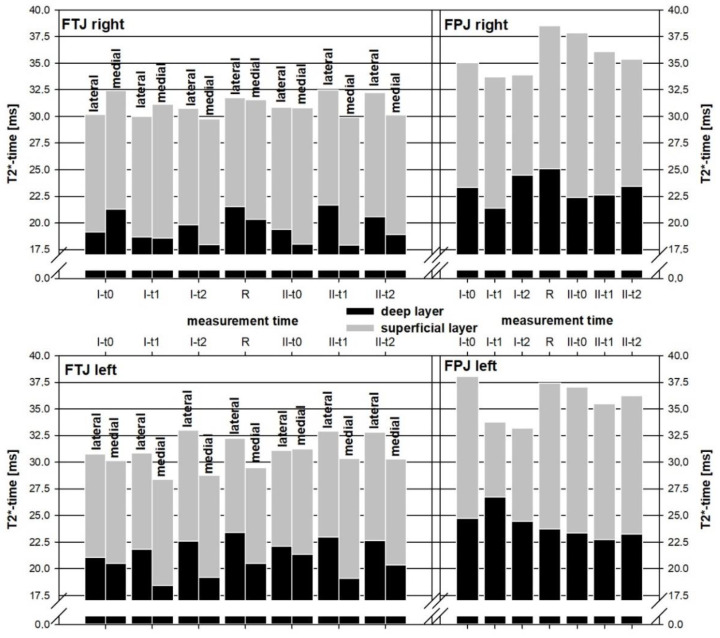
Absolute Changes of Layer-Related Mean Cartilage T2*.

**Figure 6 diagnostics-12-01391-f006:**
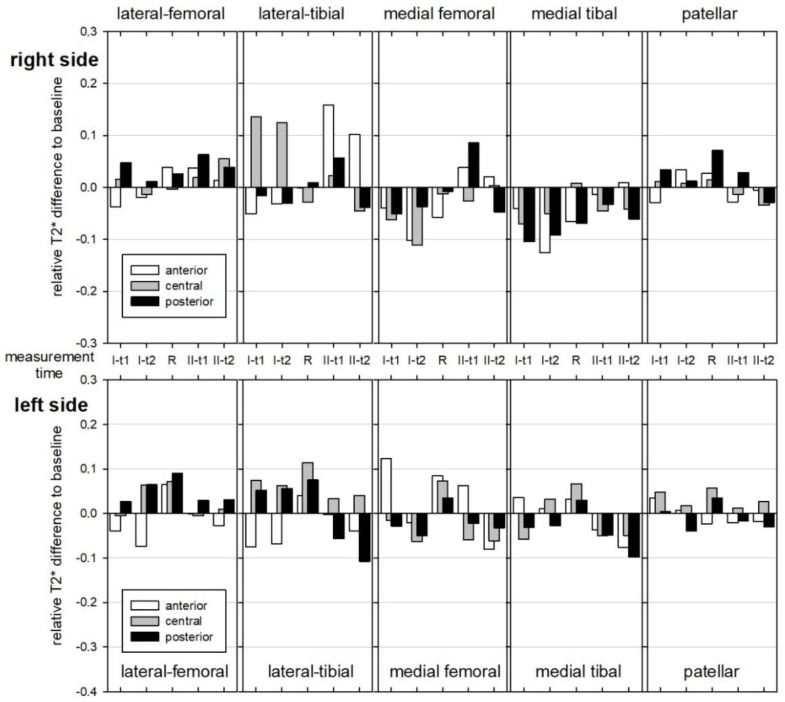
Relative Changes of Zone-Related Mean T2* Time in the Knee Joint Regions.

**Table 1 diagnostics-12-01391-t001:** MRI protocol parameters of the knee joint.

Parameter	Syngo^tm^ MapIt FLASH 2D: T2*-GRE		PDfs-TSE 2D Cor	TIRM 2D Sag
	Sag (FTJ)	Tra (FPJ)		
**FA: flip angle [°]**	60		150	140
**TE: time to echo [ms]**	4.2/12.2/19.9/27.7/35.4		31	50
**TR: repetition time [ms]**	1120	889	4400	4010
**TI: time of interval [ms]**	-			150
**ST: slice thickness [mm]**	3.0			
**FOV: field of view [mm^2^]**	289			
**MS: matrix size [pixel]**	512 × 512			256 × 256
**PS: pixel size [mm²]**	0.110			0.441
**IPR: in plane resolution [iso]**	0.332			0.664

FLASH: fast low angle shot, GRE: gradient echo, PDfs: proton density fat saturated, TIRM: turbo inversion recovery magnitude, sag: sagittal, tra: transversal, cor: coronar.

**Table 2 diagnostics-12-01391-t002:** Outline of ROIs for cartilage segments, layers, and zones in the FTJ and retropatellar.

FTJ: ROIs	Zones	Layers	Segments
**Slices in FTJ**			
(1) femoral deep-anterior	femoral anterior (1,4)	femoral deep (1,2,3)	femoral (1,2,3,4,5,6)
(2) femoral deep-central			
(3) femoral deep-posterior	femoral central (2,5)		
(4) femoral superficial-anterior		femoral superficial (4,5,6)	
(5) femoral superficial-central	femoral posterior (3,6)		
(6)femoral superficial-posterior			
(7) tibial deep-anterior	tibial anterior (7,10)	tibial deep (7,8,9)	tibial (7,8,9,10,11,12)
(8) tibial deep-central			
(9) tibial deep-posterior	tibial central (8,11)		
(10) tibial superficial-anterior		tibial superficial (10,11,12)	
(11) tibial superficial-central	tibial posterior (9,12)		
(12) tibial superficial-posterior			
**Slices in FPJ**			
(1) patellar deep-medial	patellar medial (1,4)	patellar deep (1,2,3)	patellar (1,2,3,4,5,6)
(2) patellar deep-central			
(3) patellar deep-lateral	patellar central (2,5)		
(4) patellar superficial-medial		patellar superficial (4,5,6)	
(5) patellar superficial-central	patellar lateral (3,6)		
(6) patellar superficial-lateral			

FLASH: fast low angle shot, FTJ: femorotibial joint, GRE: gradient echo, Ht: height, ROI: region of interest.

**Table 3 diagnostics-12-01391-t003:** Zonal comparison of mean intrachondral T2* time (ms) with SD in the FTJ and retropatellar.

Segments	Zonal Comparison, Mean (SD) [ms]		
	**Anterior vs. Central**	**Anterior vs. Posterior**	**Central vs. Posterior**
**FTJ lateral femoral**	−3.2 (3.9)	−6.4 (5.2)	−3.5 (2.4)
**FTJ lateral tibial**	−0.2 (1.7)	−2.6 (3.2)	−2.5 (3.1)
**FTJ medial femoral**	0.2 (4.9)	−4.8 (5.1)	−5.0 (2.9)
**FTJ medial tibial**	1.3 (1.8)	0.2 (2.2)	−1.1 (1.6)
	**Lateral vs. Central**	**Lateral vs. Medial**	**Central vs. Medial**
**Retropattelar**	1.7 (1.5)	3.9 (2.3)	2.3 (1.9)

## Data Availability

The data supporting the reported results can be found in the Appendix A. The data supporting the reported results can also be found in the hospital’s PACS system.

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
