# Peer review of "T2*-Mapping of Knee Cartilage in Response to Mechanical Loading in Alpine Skiing: A Feasibility Study"

_diagnostics, 2022, doi:10.3390/diagnostics12061391_

Round 1

Reviewer 1 Report

diagnostics-1744737-peer-review-v1

Review of T2*-mapping of knee cartilage in response to mechanical loading in alpine skiing: a feasibility study

The manuscript submitted by Schutz and co-authors describes the analysis of the impact of skiing on the knee cartilages and for this propose a protocol to image knees with MRI and assess different ski equipment. The equipment is used by 4 skiers who alternate using one type and later a different type and then the images are quantified.

In general, I found the study quite interesting, and I think that the impact of the work will quickly extend beyond the academic remit into the areas of sports and in general anyone who practices this sport and others that may have high load on joints. The study itself is robust to the level that it can be done at this stage, the sample size is small (4) and it is not feasible to blind the users of the type of equipment being used. But that is fine at this stage as it is proposed as a feasibility study.
My sole criticism is that at times the manuscript becomes difficult to read as it seems that the authors miss the fact that not everyone will be familiar with their acronyms and conventions, and thus I suggest that the manuscript will be revised with a general reader in mind. There will be several minor corrections that are necessary to make the manuscript more readable. For example:

·        * All acronyms must be defined before they are used. In the abstract FTJ is used and this is defined later in the text. ICCINTRA, CINTER, LOAs, CI, all these should be defined. Then there are the ones that are not standard, and the authors used in their study, especially in figures, e.g.  fu1-1, fu1-2, R, ACTRI-Conv. Some of these are defined, some are not, and when used in figures, the caption should explain these.

·        * Figures. Some figures are confusing and could be improved, e.g. Fig 3, why is c wider than b? The caption of 3 is just too brief, and should be used to clarify things like conv-Actri and fu-1-1 and fu 1-2. In Fig 5 the labels overlap, not only it is confusing to see fu1-1fu1-2 but overlapping makes it really hard to read and interpret. Figure 6 by contrast is very clear and perhaps this would be shown before 3 to help the interpretation of 3. Similarly in Figure 2, what are the color codes? Why is there a pair of green and a pair of red lines, what is the difference between these? What is the color map of BCD? It takes some time to correlate that the numbers in BCD match those of table 2, so it should be good to say this. In B, why is it that the red-orange region does not correspond to any number? In D there is an interesting drop of intensity towards the left of zones 1 and 4, what does this mean?

·        * Finally a few minor points, not all readers will be familiar with foehn storms, there is a space missing in line 91 after the bracket and the bold in “For” after Statistics and Testing should be removed.

Author Response

Reviewer 1: My sole criticism is that at times the manuscript becomes difficult to read as it seems that the authors miss the fact that not everyone will be familiar with their acronyms and conventions, and thus I suggest that the manuscript will be revised with a general reader in mind. There will be several minor corrections that are necessary to make the manuscript more readable. For example:

  • All acronyms must be defined before they are used. In the abstract FTJ is used and this is defined later in the text. ICCINTRA, CINTER, LOAs, CI, all these should be defined. Then there are the ones that are not standard, and the authors used in their study, especially in figures, e.g.  fu1-1, fu1-2, R, ACTRI-Conv. Some of these are defined, some are not, and when used in figures, the caption should explain these.

Response of the author: All abbreviations are defined and explained now at their first time of appearance and in the text and figures acronyms were switched in more common ones and/or explained in the captions.

  • Reviewer 1: Some figures are confusing and could be improved, e.g.

Answer of the author: The following tabs and figures were improved:

Table 1: Parameters are explained in the tab now, for better reading and easier understanding

Reviewer 1: Figure 6 by contrast is very clear and perhaps this would be shown before 3 to help the interpretation of 3.

Response of the author: Figure 6 (now 3) has been moved to the front of Figure 3 (now 4) as an example. The figures have been renumbered accordingly.

Reviewer 1: Fig 3, why is c wider than b? The caption of 3 is just too brief and should be used to clarify things like conv-Actri and fu-1-1 and fu 1-2.

Response of the author: In Figure 4 (former 3) c ist not wider than b. The relative T2*-differences are in the same range from -0.2 to +0.2!?

Reviewer 1: In Fig 5 the labels overlap, not only it is confusing to see fu1-1fu1-2 but overlapping makes it really hard to read and interpret.

Response of the author: In Figure 4 (former 3), 5 (former 4) and 6 (former 5) and in the text the abbreviation „fu“ (for follow-up) was replaced with the more common „t“ (for time). In adaption to this, the captions and figures were optimized regarding reading, interpretation and clarification.

Reviewer 1: Similarly in Figure 2, what are the color codes? Why is there a pair of green and a pair of red lines, what is the difference between these? What is the color map of BCD? It takes some time to correlate that the numbers in BCD match those of table 2, so it should be good to say this.

Response of the author: Explanation of color codes and reference to Tab.2 was implemented in the caption. The femoral part of the FPJ was not analysed due to the limits of the method regarding this small and highly different area of biomechanical load.

Reviewer 1: In B, why is it that the red-orange region does not correspond to any number?

Response of the author: According to the validated T2*-mapping method used (Mamish et al. [21]), the regions (ROIs) were drawn from border to borders of the meniscal borders, representing the central loading area of the FTJ.

Reviewer 1: In D there is an interesting drop of intensity towards the left of zones 1 and 4, what does this mean?

Response of the author: In the FTJ the lowest loading appears in the posterior zone (ROI 3,6 and 9,12), and the main loading appears in the anterior and central zones (ROIs 1,2,4,5 and 7,8,10,11). Therefore, the free cartilage water is reduced in the posterior ROIs (high signal) compared to anterior ROIs (low signal). Also, the deep layers (ROIs 1,2,3 and 7,8,9) had less compression, than the superior ones (ROIs 4,5,6 and 10,11,12)

  • Reviewer 1: Finally, a few minor points, not all readers will be familiar with foehn storms, there is a space missing in line 91 after the bracket and the bold in “For” after Statistics and Testing should be removed.

Response of the author: The term „foehn storms“ is now explained (in brackets) and space missing and bold types were corrected.

Reviewer 2 Report

The manuscript presents the results of an interesting application for MRI study of cartilage.

The research methodology is well set up, the case history good and the expected results interesting, although limited to a specific pathology.

It would be interesting if the authors would also specify other possible applications, such as in the study of degenerative or post traumatic pathology.

Author Response

Reviewer 2: The manuscript presents the results of an interesting application for MRI study of cartilage.

The research methodology is well set up, the case history good and the expected results interesting, although limited to a specific pathology.

It would be interesting if the authors would also specify other possible applications, such as in the study of degenerative or post traumatic pathology.

Response of the author: The underlying radiological method was initially developed to quantify post-traumatic and degenerative knee osteoarthritis. This has already been discussed in the "Material/Methods" chapter and is exemplarily documented with references (lines 108-123).